# Analysis and Report of the Physical and Rehabilitation Medicine Evaluation Activity in Patients Admitted to Acute Care Setting: An Observational Retrospective Study

**DOI:** 10.3390/ijerph20116039

**Published:** 2023-06-02

**Authors:** Andrea Bernetti, Marco Ruggiero, Pierangela Ruiu, Martina Napoli, Rossella D’Urzo, Annalisa Mancuso, Flavio Mariani, Luigi Tota, Francesco Agostini, Massimiliano Mangone, Marco Paoloni

**Affiliations:** 1Department of Anatomical and Histological Sciences, Legal Medicine and Orthopedics, Sapienza University, 00185 Rome, Italy; 2Physical and Rehabilitation Medicine, IRCCS Istituto Ortopedico Rizzoli, 40136 Bologna, Italy

**Keywords:** rehabilitation setting, assessment scales, PRM evaluation, disability, appropriateness

## Abstract

Background. Disability (both temporary and transitory, or definitive) might occur for the first time in a given patient after an acute clinical event. It is essential, whenever indicated, to undergo a Physical Medicine and Rehabilitation assessment to detect disability and any need for rehabilitation early. Although access to rehabilitation services varies from country to country, it should always be governed by a PRM prescription. Objective. The aim of the present observational retrospective study is to describe consultancy activity performed by PRM specialists in a university hospital in terms of requests’ typology, clinical questions, and rehabilitation setting assignment. Methods. Multiple parameters were analyzed (clinical condition, patient’s socio-family background, and rehabilitation assessment scale scores) and a correlation analysis was performed between the analyzed characteristics and both the different clinical conditions and the assigned rehabilitation setting. Results. PRM evaluations of 583 patients from 1 May 2021 to 30 June 2022 were examined. Almost half of the total sample (47%) presented disability due to musculoskeletal conditions with a mean age of 76 years. The most frequently prescribed settings were home rehabilitation care, followed by intensive rehabilitation and long-term care rehabilitation. Conclusions. Our results suggest the high public health impact of musculoskeletal disorders, followed by neurological disorders. This is, however, without forgetting the importance of early rehabilitation to prevent other types of clinical conditions such as cardiovascular, respiratory, or internal diseases from leading to motor disability and increasing costs.

## 1. Introduction

Rehabilitation consists of “a set of measures that assist individuals, who experience, or are likely to experience, disability to achieve and maintain optimal functioning in interaction with their environments” [1].

From a health system perspective, rehabilitation, together with disease prevention, health promotion, disease control, and supportive and palliative care, represents one pillar of health strategies [2].

There is an increasing need for rehabilitation worldwide due to the epidemiological shift from communicable to noncommunicable diseases, rapid global population aging, the rise in physical and mental health challenges, injuries, and comorbidities, and new rehabilitation needs related to infectious diseases such as coronavirus disease. Nevertheless, rehabilitation represents a largely unmet need globally, considering that in many countries, more than 50% of people do not receive rehabilitation services [3].

Although access to rehabilitation services varies from country to country and often within the same country, even by region, it should always be governed by a Physical Medicine and Rehabilitation (PRM) prescription [4]. This is generally based on individual clinical assessment and on a functional evaluation carried out through the use of validated scales and indicators. The adoption of standardized tools to evaluate the need for rehabilitation is important to reduce the subjectivity of clinical judgment and to make access to rehabilitation services homogeneous [1,5].

To date, the literature on the most important rehabilitative prognostic factors to be searched in order to assess the proper rehabilitation setting is scant; the few available evidence comes from experts’ opinions derived from a Delphi consensus study [6,7].

The need to identify factors affecting access to different rehabilitation settings (intensive, long-term care, and high specialization) has been the subject of several studies over the years.

Ottenbacher et al. has already postulated the relevance of “non-clinical factors” in playing a key role in decision making for PAC access. In fact, they categorize barriers to access into four categories: financial, structural, personal and sociodemographic, and attitudinal [8]. All four are defined as factors associated with differential access to PAC regardless of diagnosis, disability, or age [9].

More recently, other authors have tried to identify clinical assessment schedules (CLAS) in which functioning data are collected to guide and inform the decision-making process in clinical practice and service provision [10]. A study by Hsu YH et al. explored the factors influencing the choice of PAC services and models among stroke patients and their families through qualitative in-depth interviews [11]. 

In Europe, other nations have drawn up national documents to define pathologies requiring rehabilitation and to regulate access to rehabilitation at a national level, as in the case of France [12].

As regards Italy, appropriateness for rehabilitation should be regulated by Ministerial national guidelines which, however, are currently being tested nationwide [13]. For this reason, each region had provided its own regional guidelines. For the purposes of the present study, we referred to the Lazio Region’s guidelines [14]. 

In order to ensure an effective continuity of the rehabilitation pathway to a patient with a disability susceptible to improvement, a connection between the acute, postacute, and territorial phases is essential. In Italy, inpatient rehabilitation is divided into two areas: intensive rehabilitation and the highly specialized intensive rehabilitation in its various fields (acute brain injury, spinal cord injury, cardiologic, pulmonary, and pediatric acquired brain injury). A systematic connection among the different care settings (acute hospitalization, hospital rehabilitation, and territorial rehabilitation) guarantees a global intervention with a complete taking care of the patient, timeliness in rehabilitation treatment, and a greater appropriateness in resource allocation. The transfer from an acute care ward to a hospital or territorial rehabilitation unit must be planned through a joint assessment by the acute ward and PRM specialists; it is fundamental to perform a multidimensional evaluation based on clinical conditions, the level of disability, and relying on validated assessment scales [14].

Furthermore, to maximize the effectiveness of PRM evaluations in acute wards and to ensure early taking care of the patient, it is essential that this evaluation occurs as early as possible. Several studies in the literature have reported an association between early admission to rehabilitation care and improved functional outcomes [15,16,17].

The delay in PRM consultation may be due to multiple factors. Among them could be considered the acute ward specialist’s difficulty in timely intercepting the patient’s rehabilitation needs, the high demand, and the difficulty to provide all evaluations in a brief time.

The aim of the present observational retrospective study is to describe consultancy activity performed by PRM specialists in a university hospital, in terms of requests’ typology, clinical questions, and the final decision about rehabilitation setting assignment after an acute ward to better improve our healthcare system in terms of rehabilitation services. 

The analysis of the main parameters during PRM evaluation is aimed at standardizing as much as possible the activity carried out in view of a correct rehabilitation setting assignment.

## 2. Materials and Methods

### 2.1. Study Design

The present work is an observational retrospective study. It was conducted at the “Policlinico Umberto I” of Rome, the third largest Italian hospital with 1235 beds and approximately 41,000 hospitalizations per year. In our hospital, patients—hospitalized in acute wards—requiring rehabilitation are reported to our department through a request for PRM assessment, which is performed at acute wards by a PRM physician who provides a report on the patient’s medical record.

The study was conducted in accordance with the Helsinki Declaration (in its latest version).

We examined PRM assessments performed over a period of time ranging from 1 May 2021 to 30 June 2022 by analyzing anamnestic data, clinical and social/housing conditions, the scores of specific assessment scales, the Barthel Index [18,19,20] and CIRS [21], and the rehabilitation setting proposed for each patient. Patients with acute brain injury (ABI) and spinal cord injuries (SCIs) were assessed, respectively, according to the specific Glasgow Outcome Scale (GOS) [22], Level of Cognitive Functioning (LCF) [23], Disability Rating Scale (DRS) [24], the American Spinal Injury Association (ASIA) [25], and Spinal Cord Independence Measure (SCIM) [26] scales; data related to these assessment scales were not reported since they were too few for statistical purposes.

The number of PRM evaluation requests received by our department may vary according to the surgical activity of each department (Orthopedics above all) and it may undergo periodic variations (fewer requests in July–August), reaching an average of about 60 requests per month. Over the indicated time frame, approximately 780 PRM consultancies were requested, corresponding to 1 PRM evaluation for every 70 hospitalized patients (with the total number of annual hospitalizations being about 41,000). Departments requiring the greatest number of PRM evaluations were Neurology and Orthopedics. In our hospital, the Orthopedics department has 30 beds. As regards Neurology, however, the Nervous Diseases department is divided into different wards: Cognitive Neurorehabilitation with 8 beds, Sub-Intensive Neurology with 7 beds, Neurology with 16 beds, and the Stroke unit with 8 beds.

### 2.2. Participants

Each PRM evaluation request received by our department was considered eligible. No selection criteria were applied.

### 2.3. Characteristics and Data Sources/Measurement

The PRM physician team included medical specialists and PRM postgraduates of “Sapienza” University of Rome.

Multiple parameters that might affect the rehabilitation setting choice were analyzed, such as the clinical condition underlying acute hospitalization, patient’s socio-family background, and rehabilitation assessment scale scores. Subsequently, a correlation analysis was performed between the analyzed characteristics and both the different clinical conditions and the assigned rehabilitation setting.

Specifically, we investigated how assessment scales scores vary after acute events such as orthopedic intervention, stroke, exacerbation of an internal disease, and their distribution according to the assigned rehabilitation setting.

In addition, for each PRM evaluation, the time—in terms of number of days—elapsed between the patient’s hospitalization and the request for evaluation was assessed to analyze the priority for rehabilitation treatment in relation to the clinical condition underlying hospitalization.

### 2.4. Bias

As variability of judgment among the physicians might have arisen, PRM physicians were asked to try to minimize any observer-related bias, keeping a uniform standard of judgment. The purpose of this data collection was to precisely identify the parameters required to standardize PRM evaluations as much as possible, thereby avoiding evaluation bias.

### 2.5. Statistical Methods

Data derived from PRM evaluations were collected in a database. An observational analysis was carried out to evaluate the distribution of different characteristics (sample size, age, sex, Barthel Index, and CIRS scale) according to patients’ diagnosis (established by International Classification of Disease, ICD-9) and rehabilitation setting. The time elapsed between the day of admission to the ward and PRM evaluation request was analyzed using the median statistical index. 

## 3. Results

We analyzed data of 583 consultations performed over a period ranging from 1 May 2021 to 30 June 2022. Currently, further data are being gathered. 

We initially collected PRM evaluations carried out in “clinical condition” groups according to the Italian classification of Major Diagnostic Categories (MDC). We then analyzed the distribution of the characteristics collected in patients with disabilities of musculoskeletal (MSK), neurological, respiratory, and cardiologic origin, merging data relating to other patients suffering from internal diseases into a single category under the name of “Other” conditions.

Parameters such as the number of cases, sex, mean age, Barthel Index, and CIRS were related to the underlying clinical condition.

We analyzed 583 PRM evaluations in total (323 female and 260 male) with a female mean age of 76 years and a male mean age of 69. The mean BI was 20 for female assessments and 19 for male assessments, while CIRS was ≥3 in 48% (female) and 41% (male). In Table 1, we report the distributions of the analyzed characteristics according to disease group and divided by sex. 

Patients with MSK conditions exhibited a mean age of 76 years and a mean BI of 21, and 47% of them reported a CIRS score ≥ 3. Patients with neurological disabilities showed a mean age of 67 years and a mean BI of 15, and 37% of cases reported a CIRS score ≥ 3. Patients with respiratory disabilities displayed a mean age of 75 years with a mean BI of 25, and in 38% of cases, they reported a CIRS score ≥ 3. Patients with cardiologic disabilities presented a mean age of 79 years, a mean BI equal to 23, and a CIRS score ≥ 3 in 66% of cases. Patients suffering from internal diseases included in the “Other” category showed a mean age of 72 years, with a mean BI equal to 22 and a CIRS score ≥ 3 in 48% of cases.

Next, as reported in Table 2, by analyzing the groups of clinical conditions for which our counseling was most frequently requested (orthopedic and neurological disabilities and patients suffering from internal diseases), it emerged that the most prescribed setting among the patients with MSK disabilities was intensive rehabilitation in 35.6% of cases, followed by long-term care rehabilitation in 28.1% and home rehabilitation in 22.9%. In patients with neurological disabilities, 27.7% of patients were assigned to intensive rehabilitation, 20.6% to ABI rehabilitation, 15.6% to long-term care, 12.8% to home rehabilitation, and 5% to SCI rehabilitation. Among patients suffering from internal diseases, the most prescribed setting was home rehabilitation with 45.4%, followed by long-term care with 20.6% and intensive rehabilitation with 5.2%. In this table, the total does not correspond to 100% because we selected only the most prescribed settings and did not report settings with lower percentages (such as outpatient clinics, residential care homes, palliative care, not-definable settings, welfare services, and day hospitals).

We then analyzed the most frequently occurring diseases (according to ICD-9) during PRM evaluation activity, which are shown in order of frequency in Figure 1. It can be noticed that the most frequent clinical condition requiring consultancy was femoral neck fracture.

Patients with femoral neck fractures (Table 3) presented a mean age of 82 years and a mean BI equal to 18, and 54% of these patients reported a CIRS score ≥ 3. The mean age of patients with ischemic stroke was 73 years, with a mean BI equal to 15 and a CIRS ≥ 3 in 54% of cases. Patients in bed confinement status caused by multiple internal disorders presented a mean age of 74 years, a mean BI equal to 21, and a CIRS score ≥ 3 in 62% of cases. Patients with COVID-19 pneumonia showed a mean age of 83 years, a mean BI of 24, and a CIRS ≥ 3 in 46% of cases. Finally, patients with intracerebral hemorrhage reported a lower mean age (64 years), a mean BI equal to 10, and a CIRS ≥ 3 in 28% of cases.

This analysis was carried out only for the most frequently occurring clinical conditions (>5 consultations); therefore, the cases shown in Table 3 are lower than the total consultations performed (294 vs. 583).

The proposed rehabilitation setting in relation to each clinical condition underlying hospitalization is depicted in Table 4. With regard to home rehabilitation, 35% of patients were affected by MSK diseases, 11% were suffering from neurological diseases, 22% were affected by respiratory diseases, 8% were affected by cardiovascular diseases, and 24% affected by internal diseases (Other). In intensive rehabilitation units, as well as in long-term care, the majority of patients (64% and 55%, respectively) were affected by musculoskeletal conditions.

The most frequently assigned rehabilitation setting (Table 5) was home rehabilitation, being assigned in 168 evaluations out of 570. The mean age of patients assigned to this setting was 78 years, with a mean BI equal to 26 and a CIRS ≥ 3 in 36% of cases. Intensive rehabilitation was the second most frequent setting, assigned to 141 patients: their mean age was 68 years, with mean BI equal to 25 and a CIRS ≥ 3 in 21% of cases. The third most frequent rehabilitation setting was long-term care rehabilitation, assigned in 130 evaluations. The mean age of patients assigned to this setting was 81 years, with a mean BI equal to 13 and a CIRS score ≥ 3 in the majority of cases (97%). Other settings followed, as depicted in Table 4.

In Table 6, the number of cases related to the main clinical conditions found during our PRM evaluations (assessed using ICD-9) is reported according to the proposed rehabilitation setting. The clinical conditions most frequently assigned to intensive rehabilitation were femoral neck fracture and ischemic stroke, similarly to those most frequently assigned to long-term care rehabilitation. This analysis was carried out only for the most frequently occurring clinical conditions (>5 consultations); therefore, the cases shown in Table 6 are lower than the total consultations performed.

The median number of days elapsed between the admission of each patient to the acute ward and the request for PRM evaluation is shown in Table 7.

It can be noticed that the time was longer in the category of patients with cardiovascular disabilities (22.5 days), followed by patients with respiratory diseases (20.5 days). On the other hand, it was shorter for patients with MSK diseases (6 days), neurological patients (12 days), and for the ones in the “Other” category (18 days). Since it was not possible to obtain these data for all the sampled patients, the total number of cases reported in the table below is lower than the total number of patients included in the present study (516 vs. 583). 

## 4. Discussion

In order to assess quality levels, the timeliness of the rehabilitation treatments, and the appropriateness of access for various conditions, we analyzed consultation requests received by the PRM department of “Policlinico Umberto I” Hospital.

Data related to 583 PRM evaluations were collected and analyzed retrospectively, over a timeframe ranging from 1 May 2021 to 30 June 2022.

It should be emphasized that this study took place during the pandemic period caused by COVID-19, which had repercussions on acute care activity and, consequently, on rehabilitation [27,28]. Due to Policlinico Umberto I being a COVID-19 hospital, rehabilitation services have not been closed as has happened in other hospitals, but rather, the effort has been given to hospitalized patients and rehabilitation services have worked mainly for patients in the transition from the acute to postacute phase.

Most of the PRM evaluations were carried out on patients with disabilities of MSK origin, followed by those with disabilities of neurological origin; this suggests that these clinical conditions are still among those of greatest interest in rehabilitation. PRM evaluations have been carried out in multiple internal medicine departments and in intensive care units [29,30]. The volume of activities recorded in relation to these departments reveals a key role of the PRM physician, involved since the acute phase [5,31]. In fact, any given clinical condition, throughout its course, can cause disability for the patient, thus determining the need for a rehabilitation evaluation for appropriate management.

The analysis of the collected parameters—including the number of cases, average age, BI, and CIRS by clinical condition—shows that the largest patient group was composed of subjects suffering from MSK conditions, with a mean age of 76 years. These patients presented, on average, a lower BI compared to patients with cardiovascular and respiratory diseases. Conversely, patients with neurological disorders had a lower mean age and, compatibly with the consequences that may result from an event such as a stroke or a brain trauma, they showed a mean BI lower than other categories, scoring a BI between 0 and 24 in almost 75% of cases. Although the epidemiology of brain ischemia and cardiovascular diseases is comparable, it has to be noted that the “neurological disorders” category also includes head and spinal traumas, which generally tend to affect young subjects.

We then analyzed the single clinical conditions most frequently encountered during the PRM consultations. MSK and neurological disorders were the most frequent, with femoral neck fractures [32] being the clinical condition for which we were most frequently asked for an evaluation, followed by ischemic stroke, hemorrhagic stroke, bed confinement status for internal diseases, COVID-19-related pneumonia [33], and subarachnoid hemorrhage, coherently with the above-mentioned data sorted by clinical condition. Patients affected by COVID-19-related pneumonia had a higher mean age and were affected by three or more comorbidities in 46% of cases, thus confirming the SARS-CoV-2 virus’s tendency to determine more severe clinical pictures in immunocompromised patients who are of an older age with comorbidities [26,27]. From this analysis, the epidemiological and socioeconomic impact of femoral neck fracture and the need to adopt pharmacological and nonpharmacological strategies to prevent this condition and its related disability emerged [34].

As regards the settings, the most frequently proposed were home rehabilitation in 168 cases, intensive rehabilitation in 141 cases, and long-term care in 130 cases. Patients assigned to intensive rehabilitation presented a lower age than patients assigned to home rehabilitation [35] or long-term care. Intensive rehabilitation setting patients are required to sustain three hours of rehabilitation therapy per day; in cases where the ability to cope with such effort is limited, due to increasing age or comorbidities, it is considered preferable to propose different and more adequate settings from which the patient would benefit most and which would allow for limiting the probability of developing complications and infections that may occur in a hospital environment.

It is worth presenting the data gathered on the time elapsed between the hospitalization in ward and the request for PRM consultation [36]. Remarkably, in patients with respiratory and cardiovascular disabilities, a median time of 20.5 and 22.5 days, respectively, was recorded. Such a significant delay could be ascribed to the time required to reach clinical stabilization in these patients and to the fact that hospitalization in acute wards was generally preceded by intensive/resuscitation care. Moreover, in these cases, the long time between hospitalization and PRM evaluation may consequently result in a progressive loss of functional independence in hospitalized patients, therefore forcing them into prolonged periods of bed rest. In patients suffering from MSK conditions, a median time of 6.0 days was scored from the day of admission to the PRM evaluation. In this regard, considering that, in most cases, patients suffer from time-dependent diseases—in particular, femoral neck fractures—such a time interval should be further shortened in order to ensure rapid access to rehabilitation treatment. A median time of about 12 days was reported in patients with neurological disabilities, indicating the need for a longer time for clinical stabilization compared to patients with orthopedic disabilities. 

The overall findings of the present report firstly indicate that PRM counseling is crucial to ensure the appropriateness of rehabilitation programs. Secondly, it emerges that the PRM specialist is the only medical specialist able to globally take care of the patient with disabling outcomes of any clinical condition and along the entire course of the disease, from the acute phase [5] to the following territorial phases.

In fact, the PRM specialist provides consultation as early as possible from the acute event and, if there are indications for rehabilitation treatment, performs diagnoses and defines rehabilitation programs. Additionally, the PRM specialist globally evaluates the patient’s clinical, functional, and social conditions and the degree of compliance with rehabilitative treatment, identifying the most suitable rehabilitation pathway and the most appropriate settings, taking into account the rehabilitation intensity and the healthcare support needed by the patient. Hence, the patient undergoes a holistic examination, no longer focusing only on the acute problem leading to hospitalization but also deeply evaluating the social and environmental factors, in order to prevent hospitalization and to promote home stay [37].

A reduction in the time between hospitalization and PRM consultation request may be highly advantageous in order to ensure high quality and timeliness in their rehabilitation treatment and to minimize the rate of inappropriate access [38,39]. Indeed, delayed access to rehabilitation determines a cascade effect: patients with late PRM evaluations occupy, on average, rehabilitation beds longer, resulting in further delays for prospective patients. These delays may also be associated with patients’ negative functional scores at both admission and discharge. In the future, it could be interesting to analyze the correlation between functional scales’ scores and waiting times for access to rehabilitation [38].

The definition of the rehabilitation process must consider different intervention settings (hospital, outpatient treatment, and home treatment) based on the assessment of the patient’s overall function and needs. From this perspective, the design and implementation of the proposed rehabilitation interventions are not only based on the clinical condition; rather, they need to be shaped according to the individuality of patients and to the contexts in which they express their social functioning. In this regard, one of the strategies to be mentioned is telerehabilitation, an innovative approach which ensures rehabilitative intervention even in home settings, especially in the pandemic period [40].

From the analysis carried out, it emerged that the most frequently proposed setting for orthopedic and neurological rehabilitation was intensive rehabilitation, regardless of the clinical condition causing the disabling outcomes. In contrast, with regard to internal disease and respiratory or cardiological diseases, the assigned setting was home rehabilitation.

The strengths of the present study are that it is original, innovative, and it develops a topic not previously analyzed in the literature. It presents a large and heterogeneous sample of patients, with a relevant variety of clinical conditions included.

Some limitations must be described. Being a retrospective observational study, it analyzes the entire sample without applying selection criteria to the patient pool. It is not a multicentric study. The results have not been compared with other hospitals’ data; nevertheless, it has to be considered that “Policlinico Umberto I” Hospital of Rome is a reference center, presenting a wide spectrum of clinical conditions to be analyzed. Another limitation of this study is the lack of data on the patients’ follow-up for evaluating the effectiveness of the proposed settings. However, although it is a local study, it clearly shows that in the rehabilitation field, the treatment pathway and the individual rehabilitation project are not defined based only on the impairment; rather, they are designed according to the rehabilitation diagnosis related to the patient’s function and needs.

This is the first Italian study conducted on a large scale in relation to physiatric medical consultations. Having been conducted in one of the largest hospitals in Italy, it is likely to expect that these results could be useful at an international level to better understand the type of patients who need physiatric consultation and what are the specific characteristics that guide the physiatrist in expressing the rehabilitative diagnosis and prognosis. The present research could also provide general indications for the purpose of resource planning and allocation, to be defined on the basis of the analysis of epidemiological data of patients admitted to the structures of interest.

## 5. Conclusions

PRM assessments performed for acute care inpatients are very important in several respects. Firstly, it should be remembered that rehabilitation diagnosis is pivotal for determining the patient’s rehabilitation treatment pathway, starting from the very first stages of the clinical condition for which the patient is hospitalized. Secondly, early definition of the individual rehabilitation project allows one to quickly identify the most appropriate rehabilitation setting for each patient from the time of discharge from the hospital ward. It is crucial to ensure the timely global taking care of the patient, through an early collaboration between the acute ward and the PRM specialists, to improve the outcome and reduce disability.

## 6. Patents

There are no patents resulting from the work reported in this manuscript.

## Figures and Tables

**Figure 1 ijerph-20-06039-f001:**
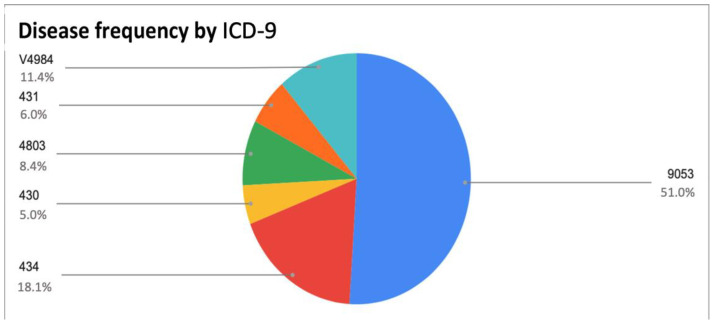
Frequencies of diseases as evaluated using ICD-9. The relevant codes are shown, namely, 9053 (late effect of fracture of neck of femur), 434 (cerebral thrombosis), V4984 (bed confinement status), 4803 (pneumonia due to SARS-associated coronavirus), 431 (intracerebral hemorrhage), and 430 (subarachnoid hemorrhage).

**Table 1 ijerph-20-06039-t001:** Distribution of the analyzed characteristics (number, sex, mean age, mean BI, mean standard deviation BI, and CIRS ≥ 3 scores) according to disease group. M = male, F = female.

Pathology	N° of Cases	Sex	Mean Age	Mean BI	Mean St. Dev. BI	CIRS ≥ 3 (%)
MSK	253	F: 176	79	19	14	50
		M: 78	70	24	14	40
		Total	76	21	14	47
Neurologic	141	F: 63	72	16	19	38
		M: 78	63	14	15	36
		Total	67	15	17	37
Respiratory	63	F: 32	76	25	23	44
		M: 31	73	24	20	32
		Total	75	25	22	38
Cardiologic	35	F: 13	79	23	17	62
		M: 22	80	24	12	68
		Total	79	23	14	66
Other	91	F: 40	72	21	21	52
		M: 51	72	24	20	45
		Total	72	22	20	48
Values over total	583	F: 323	76	20	17	48
		M: 260	69	21	17	41
		Total	73	20	17	45

**Table 2 ijerph-20-06039-t002:** Percentage distribution of rehabilitation settings according to clinical condition groups most frequently requested during Physical Medicine and Rehabilitation (PRM) consultancy activity.

Clinical Condition	Intensive Rehabilitation	Long-Term Rehabilitation	Home Rehabilitation	ABI Rehabilitation	SCI Rehabilitation	Total
MSK (%)	35.6	28.1	22.9	/	/	86.6
Neurologic (%)	28.1	15.6	12.8	20.6	5.2	82.3
Other (%)	5.2	20.6	45.4	/	/	71.2

**Table 3 ijerph-20-06039-t003:** Distribution of the analyzed characteristics (number, mean age, mean BI, mean standard deviation BI, and CIRS ≥3 scores) according to the clinical condition assessed using ICD-9.

ICD-9 Disease	N° of Cases	Mean Age	Mean BI	Mean St. Dev. BI	CIRS ≥ 3 (%)
9053	151	82	18	12	54
434	54	73	15	14	54
V4984	34	74	21	20	62
4803	24	83	24	21	46
431	18	64	10	9	28
430	13	74	10	13	23
Values over total	294	78	17	14	51

**Table 4 ijerph-20-06039-t004:** Percentage distribution of disease groups according to the rehabilitation settings assigned during the PRM consultancy activity.

	Clinical Condition				
Rehabilitation Settings	Msk	Neurologic	Respiratory	Cardiologic	Other
Home rehabilitation	35	11	22	8	24
Intensive rehabilitation	64	28	3	2	3
Long-term care	55	17	5	8	15
Acute brain injury (ABI) rehabilitation	–	97	3	–	–
Rehab in acute wards	22	22	4	9	43
Not definable	47	26	10	4	13
Values over total	46	23	11	5	15

**Table 5 ijerph-20-06039-t005:** Distribution of the analyzed characteristics (number, mean age, mean BI, mean standard deviation BI, and CIRS ≥3 scores) according to rehabilitation settings proposed during the PRM consultancy activity.

Rehabilitation Settings	N° of Cases	Mean Age	Mean BI	Mean St. Dev. BI	CIRS ≥ 3 (%)
Home rehabilitation	168	78	26	18	36
Intensive rehabilitation	141	68	25	12	21
Long-term care	130	81	13	8	97
Acute brain injury (ABI) rehabilitation	30	57	3	5	27
Rehabilitation in acute wards	23	68	20	26	39
Spinal cord injury rehabilitation	8	61	15	16	0
Residential care home	7	76	10	15	14
Outpatient clinic	5	77	57	37	20
Welfare services	5	78	3	3	60
Day hospital	4	52	56	12	50
Hospice	2	48	9	2	100
Not definable	47	70	12	13	40
Values over total	570	74	20	16	46

**Table 6 ijerph-20-06039-t006:** Number of cases according to proposed rehabilitation setting and clinical condition using ICD-9. The relevant codes are shown, namely, 9053 (late effect of fracture of neck of femur), 434 (cerebral thrombosis), V4984 (bed confinement status), 4803 (pneumonia due to SARS-associated coronavirus), 431 (intracerebral hemorrhage), and 430 (subarachnoid hemorrhage).

	ICD-9 Disease						
Rehabilitation Setting	9053	434	V4984	4803	431	430	Total
Home rehabilitation	33	7	17	18	1	–	76
Intensive rehabilitation	51	21	–	1	4	2	79
Long-term care	52	15	8	5	2	2	84
Acute brain injury (ABI) rehabilitation	–	7	–	–	4	5	16
Not definable	12	3	3	–	5	2	25
Total	148	53	28	24	16	11	280

**Table 7 ijerph-20-06039-t007:** Time (median) elapsed between hospitalization in acute care ward and request for Physical Medicine and Rehabilitation (PRM) consultation according to the clinical condition.

Clinical Condition	N° of Cases	Median Number of Days from Hospitalization
MSK	221	6
Neurologic	129	12
Respiratory	56	20.5
Cardiologic	33	22.5
Other	77	18
Median of days from hospitalization out of the total	516	11

## Data Availability

The datasets generated during the current study are available from the corresponding author on reasonable request.

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
