# Peer review of "Analysis and Report of the Physical and Rehabilitation Medicine Evaluation Activity in Patients Admitted to Acute Care Setting: An Observational Retrospective Study"

_ijerph, 2023, doi:10.3390/ijerph20116039_

Round 1
Reviewer 1 Report
The manuscript entitled "Analysis and report of the Physical and Rehabilitation Medicine evaluation activity in patients admitted to acute care setting. An observational retrospective study" provides the observational retrospective study to describe the consultancies activity of PRM specialists.
I present my comments and suggestions for changes in relation to the following parts of the article.
Abstract
(Line 11-15) The sentence in that position is the same as the sentence mentioned in the introduction (Line 42-46). Please correct it.
(Line 19) ... from 1.05.2021 to 30.06.2022, ... I think it would be better to correct it to "from May 1, 2021 to June 30, 2022".
(Line 20) ... CIRS' scores. Please include the full terms before using an abbreviation.
Introduction
The introduction explaining the purpose of this study is too short. I suggest that you add relevant references and content to aid readers' understanding and improve readability.
Materials and Methods
(Line 83-84) GOS, LC, DRS, ASIA, SCIM... Please include the full terms before using an abbreviation.
(Line 128) ... by ICD-9... Please include the full terms before using an abbreviation.
Results
(Line 153-164) For better understanding, it would be nice to add a table or figure to easily check the results, such as Table 1 or Figure 2.
(Line 205) In Table 3, the percentage distributions for disease groups should sum to 100, but for Home Rehabiltation and Acute Rehabiltation, the percentage distributions sum to 101. Please check again and correct it.
(Line 205) (Table 3) ABI. Please include the full terms before using an abbreviation.
(Line 219) (Table 4) RSA. Please include the full terms before using an abbreviation.
(Line 231) (Table 6) MFR. Please include the full terms before using an abbreviation.
Author Response
Response to Reviewer 1 Comments
The manuscript entitled "Analysis and report of the Physical and Rehabilitation Medicine evaluation activity in patients admitted to acute care setting. An observational retrospective study" provides the observational retrospective study to describe the consultancies activity of PRM specialists.
I present my comments and suggestions for changes in relation to the following parts of the article.
Abstract
(Line 11-15) The sentence in that position is the same as the sentence mentioned in the introduction
(Line 42-46). Please correct it.
Response 1: dear reviewer, thank you for the advice. We correct the sentence as indicated.
(Line 19) ... from 1.05.2021 to 30.06.2022, ... I think it would be better to correct it to "from May 1, 2021 to June 30, 2022".
Response 2: Thank you for the suggestion. We correct the sentence as suggested.
(Line 20) ... CIRS' scores. Please include the full terms before using an abbreviation.
Response 3: Thank you for the suggestion. We add the full term “Cumulative Illness Rating Scale”as indicated.
Introduction
The introduction explaining the purpose of this study is too short. I suggest that you add relevant references and content to aid readers' understanding and improve readability.
Response 4: We improved the introduction section explaining the national guidelines in terms of rehabilitation services and the need for early rehabilitation to guarantee better clinical outcome. We have added references about early rehabilitation to improve the patient's outcome. Our retrospective study may help to understand the strengths and the limitations of rehabilitation services.
Materials and Methods
(Line 83-84) GOS, LC, DRS, ASIA, SCIM... Please include the full terms before using an abbreviation.
Response 5: We add the full terms as suggested.
(Line 128) ... by ICD-9... Please include the full terms before using an abbreviation.
Response 6: The full term was added as suggested.
Results
(Line 153-164) For better understanding, it would be nice to add a table or figure to easily check the results, such as Table 1 or Figure 2
Response 7: Thank you for the advice, a new table has been added: Table 2. In table 2 it is reported the percentage distribution of rehabilitation settings most frequently prescribed according to clinical condition groups most frequently requested during PRM consultancy activity. The total does not correspond to 100%, because we select only the most prescribed settings, not reporting
settings with lower percentages (such as outpatient clinic, residential care home, palliative care, not definable, welfare services, day hospital hospital).
1
(Line 205) In Table 3, the percentage distributions for disease groups should sum to 100, but for Home Rehabiltation and Acute Rehabiltation, the percentage distributions sum to 101. Please check again and correct it.
Response 8: Thank you for the advice, we reviewed and corrected the typo.
(Line 205) (Table 3) ABI. Please include the full terms before using an abbreviation.
Response 9: The full term was added as suggested.
(Line 219) (Table 4) RSA. Please include the full terms before using an abbreviation.
Response 10: Thank you for the advice, we correct the error with the italian abbreviation “RSA” into the international denomination residential care home.
(Line 231) (Table 6) MFR. Please include the full terms before using an abbreviation.
Response 11: Thank you for the advice, we correct the error with the abbreviation “PRM”, meaning Physical Medicine and Rehabilitation.

Reviewer 2 Report
- Abstract:
1- I suggest exchanging "pathology" and " pathologies" for clinical condition,
2- Set CIRS.
- Introduction
1- Enter any ref in line 42-46.
2- This observational study was performed during the pandemic period caused by Covid-19. I suggest that this be pointed out at some point in the description of the study and how this event may have influenced the evaluation of clinical dysfunctions.
3- It is necessary to include information on the conduct of the study having been in accordance with the Helsinki Declaration.
4- Although the authors declared that ethical issues were not mandatory, was consent to participate in the study requested? Were the participants aware that their data was being analyzed? I suggest making a brief comment on this. Even if there were any cases of refusal to participate in the study.
5- Must include a caption for table 1 and table 4.
6- I suggest that throughout the text, wherever "pathology" is, it should be replaced by clinical condition or dysfunctions. Etymologically, we attributed the term pathology to the study of the disease.
7- What rehabilitation modalities have gained prominence in the period of the pandemic caused by Covid-19 for each type of dysfunction? These recommendations can be presented at the end of the discussion, such as the strength of this work.
Author Response
Response to Reviewer 2 Comments
- Abstract:
1- I suggest exchanging "pathology" and " pathologies" for clinical condition,
Response 1: dear reviewer, we thank you for the suggestion. We integrated your suggestion along
the paper.
2- Set CIRS.
Response 2: thank you for the advice. We add a full term corresponding to the abbreviation.
- Introduction
1- Enter any ref in line 42-46.
Response 1: We improved the introduction section, removing the sentences corresponding to these
lines (this part has been moved into the abstract); we moreover explained the national guidelines in
terms of rehabilitation services and the need for early rehabilitation to guarantee better clinical
outcome. We have added references about early rehabilitation to improve the patient's outcome.
2- This observational study was performed during the pandemic period caused by Covid-19. I
suggest that this be pointed out at some point in the description of the study and how this event
may have influenced the evaluation of clinical dysfunctions.
Response 2: Dear reviewer, we thank you for the suggestion. We point out this aspect in the
discussion. See response 7.
3- It is necessary to include information on the conduct of the study having been in accordance with
the Helsinki Declaration.
Response 3: we thank you for the advice. We include the information requested.
4- Although the authors declared that ethical issues were not mandatory, was consent to participate in the study requested? Were the participants aware that their data was being analyzed? I suggest making a brief comment on this. Even if there were any cases of refusal to participate in the study.
Response 4: Thank you for the question. For our study, data were collected retrospectively and described anonymously, and the study took place at a university medical center. All patients were aware of being admitted to this center and signed their consent to data processing at the time of admission in the ward. No data as described identifies an individual patient. Anyway, if requested, we can provide a statement from the department Head acknowledging the above information.
5- Must include a caption for table 1 and table 4.
Response 5: Captions were added both for table 1 and table 4.
6- I suggest that throughout the text, wherever "pathology" is, it should be replaced by clinical
condition or dysfunctions. Etymologically, we attributed the term pathology to the study of the
disease.
Response 6: see answer to point 1.
7- What rehabilitation modalities have gained prominence in the period of the pandemic caused by Covid-19 for each type of dysfunction? These recommendations can be presented at the end of the discussion, such as the strength of this work.
Response 7: Dear reviewer, thank you for your observation. We integrated the discussion as follows: “It should be emphasized that this study took place during the pandemic period caused by Covid-19 which had repercussions on acute care activity and consequently on rehabilitation [19].
Being Policlinico Umberto I a Covid hospital, rehabilitation services have not been closed as has happened in other hospitals, but rather the effort has been given to hospitalized patients and rehabilitation services have worked mainly for patients in the transition from acute to post-acute
phase”.

Reviewer 3 Report
The researchers aimed to describe the consultancies activity of PRM specialists in a University Hospital, in terms of typology of the requests, clinical questions, and final decision about assignment of rehabilitation setting after acute inward or not.
The manuscript has errors and requires corrections by the investigators, the comments are shown below:
Comment 1º: The work requires an English review by an expert; please attach certification of the English review.
Comment 2º: The references in the text and in the methodology section do not comply with the author guidelines of the journal.
Comment 3º: Reformulate and rewrite the abstract with the following sections, they should be written in the form of: Background. This is a concise statement of the reasons for conducting this research, placing it in the context of current knowledge or controversies. Objective. This is a clear statement of the precise objective or question being addressed in the paper this may take the form of objectives or contrasting hypotheses. Methods. The basic design of the study and its duration should be described for quantitative and qualitative research quantitative and qualitative the methods used should be indicated and the data/statistical methods should be provided. Results. The main results of the study should be stated in narrative form any measurements or other information that may require explanation information that may require explanation confidence intervals are preferred to p-values values confounders, modifiers or mediators, as well as any other factors crucial to the outcome of the study should be indicated crucial to the outcome of the study should be indicated. Conclusions. The conclusions of the study that are directly supported by the evidence presented should of the evidence reported, along with clinical application, and speculation about the potential impact on current thinking current thinking.
Comment 4º: You should use keywords, mainly Mesh or DECs, which meaningfully represent your manuscript. The terms "rehabilitation" "rehabilitation services" "PRM evaluation" are too similar.
Comment 5º: The introduction is very scarce in content and does not contextualize the problem. The number of references used is very scarce and the importance of knowing (number of queries, type of queries, type of requests, factors that condition income) should be contextualized.
Comment 6º: This problem is global or only affects Italy. Justify.
Comment 7º: Delete the sentence "In our opinion, [...] The introduction serves to contextualize and expose the objective of the study, it is not indicated to make personal opinions.
Comment 8º: The introduction must be justified and contextualized; there are sections which do not present any type of justification with the current and relevant literature. See page 2 line 42-46 and line 49-53. Rephrase your introduction.
Comment 9º: It does not present approval from an ethics committee and does not follow ethical guidelines in research since it is dealing with patient data.
Comment 10º: It should describe in more detail the criteria for selecting participants so that the study can be reproducible.
Comment 11º: The variables section is wrong; it should clearly define the following variables: response, exposures, predictors, confounders and effect modifiers.
Comment 12º: For each variable of interest, provide the data sources and details of the assessment (measurement) methods, reference with scientific literature the measurement instrument used.
Comment 13º: How the sample size was determined in a subsection.
Comment 14º: Explain how quantitative variables were treated in the analysis in a subsection.
Comment 15º: A detailed description of the anthropometric characteristics of the participants should be provided, only age and pathology (not specified).
Comment 16º: Describe the number of triggering events or provide a summary and, if appropriate, measures adjusted for confounders and their precision (e.g., 95% confidence intervals). Specify the confounders for which adjustment is made and the reasons for including them (if categorizing continuous variables, describe the limits of the intervals; if appropriate, consider accompanying relative risk estimates with absolute risk estimates for a relevant time period), with reference to the variables of analysis.
Comment 17º: The discussion section should further elaborate on the clinical repercussions that may arise as a consequence of their results.
Comment 18º: In the discussion section, a more in-depth analysis of the topic of study should be carried out, making a comparison with previous studies on the topic of study.
Comment 19º: Discussions should cover the key findings of the study: discuss any previous research related to the topic to place the novelty of the discovery in the appropriate context, discuss possible shortcomings and limitations in its interpretations, discuss its integration into the current understanding of the problem and how. This advances current views, speculates on the future direction of research, and freely postulates theories that could be tested in the future, completed, and reformulated. The discussion should be rewritten to present serious errors.
Comment 20º: Move the last two paragraphs of the conclusions to the discussion section.
Comment 21º: It does not present approval from an ethics committee and does not follow ethical guidelines in research since it is dealing with patient data.
Comment 22º: The number of references is scarce for a subject as important as the one addressed by the authors.
Author Response
Response to Reviewer 3 Comments
The researchers aimed to describe the consultancies activity of PRM specialists in a University Hospital, in terms of typology of the requests, clinical questions, and final decision about assignment of rehabilitation setting after acute inward or not.
The manuscript has errors and requires corrections by the investigators, the comments are shown below:
Comment 1º: The work requires an English review by an expert; please attach certification of the English review.
Response 1: Dear reviewer, we thank you for the suggestion. The work has entirely been reviewed by an English native speaker.
Comment 2º: The references in the text and in the methodology section do not comply with the author guidelines of the journal.
Response 2: Thank you for the advice. Reference number has been increased.
Comment 3º: Reformulate and rewrite the abstract with the following sections, they should be written in the form of: Background. This is a concise statement of the reasons for conducting this research, placing it in the context of current knowledge or controversies. Objective. This is a clear statement of the precise objective or question being addressed in the paper this may take the form
of objectives or contrasting hypotheses. Methods. The basic design of the study and its duration should be described for quantitative and qualitative research quantitative and qualitative the methods used should be indicated and the data/statistical methods should be provided. Results. The main results of the study should be stated in narrative form any measurements or other
information that may require explanation information that may require explanation confidence intervals are preferred to p-values values confounders, modifiers or mediators, as well as any other factors crucial to the outcome of the study should be indicated crucial to the outcome of the study should be indicated. Conclusions. The conclusions of the study that are directly supported by the evidence presented should of the evidence reported, along with clinical application, and speculation about the potential impact on current thinking current thinking.
Response 3: Thank you for the suggestion. The abstract has been reformulated and rewritten, following your indications and with the addition of the indicated sections.
Comment 4º: You should use keywords, mainly Mesh or DECs, which meaningfully represent your manuscript. The terms "rehabilitation" "rehabilitation services" "PRM evaluation" are too similar.
Response 4: Keywords have been modified as suggested.
Comment 5º: The introduction is very scarce in content and does not contextualize the problem. The number of references used is very scarce and the importance of knowing (number of queries, type of queries, type of requests, factors that condition income) should be contextualized.
Response 5: Thank you for your observation. The introduction has been modified and integrated as suggested: We improved the introduction section explaining the national guidelines in terms of rehabilitation services and the need for early rehabilitation to guarantee better clinical outcome. We
have added references about early rehabilitation to improve the patient's outcome.
Comment 6º: This problem is global or only affects Italy. Justify.
Response 6: In this study we have analyzed the Italian situation; our retrospective study may help to understand the strengths and the limitations of rehabilitation services. Nevertheless such analysis could be applied worldwide. We also add some new references in this regard.
Comment 7º: Delete the sentence "In our opinion, [...] The introduction serves to contextualize and expose the objective of the study, it is not indicated to make personal opinions.
Response 7: Thank you for the advice, the sentence was deleted.
Comment 8º: The introduction must be justified and contextualized; there are sections which do not present any type of justification with the current and relevant literature. See page 2 line 42-46 and line 49-53. Rephrase your introduction.
Response 8: Thank you for your observation. The introduction has been modified as suggested. References were added.
Comment 9º: It does not present approval from an ethics committee and does not follow ethical guidelines in research since it is dealing with patient data.
Response 9: For our study, data were collected retrospectively and described anonymously, and the study took place at a university medical center. All patients were aware of being admitted to this center and signed their consent to data processing at the time of admission in the ward. No data as
described identifies an individual patient. Anyway, if requested, we can provide a statement from the department Head acknowledging the above information.
Comment 10º: It should describe in more detail the criteria for selecting participants so that the study can be reproducible.
Response 10: Being an observational study, we restricted ourselves to perform a retrospective analysis. All patients for whom PRM evaluation was requested were included.
Comment 11º: The variables section is wrong; it should clearly define the following variables: response, exposures, predictors, confounders and effect modifiers.
Response 11: The term “variable” can be confusing; it has been replaced with the term “characteristics”.
Comment 12º: For each variable of interest, provide the data sources and details of the assessment (measurement) methods, reference with scientific literature the measurement instrument used.
Response 12: The term “variable” has been replaced with the term “characteristics”. All characteristics have been collected and validated and standardized international scales have been used. References have been added as requested.
Comment 13º: How the sample size was determined in a subsection.
Response 13: We have not determined sample size. Consecutive data have been collected.
Comment 14º: Explain how quantitative variables were treated in the analysis in a subsection.
Response 14: The term “variable” has been replaced with the term “characteristics”. A correlation analysis was performed between the analyzed characteristics and both the different clinical conditions and the assigned rehabilitation setting.
Comment 15º: A detailed description of the anthropometric characteristics of the participants should be provided, only age and pathology (not specified).
Response 15: Thank you for the suggestion; anthropometric characteristics (sex) of the participants have been added, both in the text and in the table.
Comment 16º: Describe the number of triggering events or provide a summary and, if appropriate, measures adjusted for confounders and their precision (e.g., 95% confidence intervals). Specify the confounders for which adjustment is made and the reasons for including them (if categorizing continuous variables, describe the limits of the intervals; if appropriate, consider accompanying
relative risk estimates with absolute risk estimates for a relevant time period), with reference to the variables of analysis.
Response 16: dear reviewer, thanks for your observation. However ours is a retrospective observational study in which we analyzed the characteristics of single patients and their correlation with respect to their clinical conditions and the proposed rehabilitation setting defined after rehabilitation consultations.
Comment 17º: The discussion section should further elaborate on the clinical repercussions that may arise as a consequence of their results.
Response 17: see response 16.
Comment 18º: In the discussion section, a more in-depth analysis of the topic of study should be carried out, making a comparison with previous studies on the topic of study.
Response 18: thank you for the suggestion, the discussion section has been improved analyzing the topic more in depth. It was not possible to make a comparison with previous studies since similar articles are not present in literature to our knowledge.
Comment 19º: Discussions should cover the key findings of the study: discuss any previous research related to the topic to place the novelty of the discovery in the appropriate context, discuss possible shortcomings and limitations in its interpretations, discuss its integration into the current understanding of the problem and how. This advances current views, speculates on the future
direction of research, and freely postulates theories that could be tested in the future, completed, and reformulated. The discussion should be rewritten to present serious errors.
Response 19: Thank you for your suggestion. The discussion has been integrated including strengths and limitations of the study and the topic has been contextualized.
Comment 20º: Move the last two paragraphs of the conclusions to the discussion section.
Response 20: The last two paragraphs have been moved to the discussion section.
Comment 21º: It does not present approval from an ethics committee and does not follow ethical guidelines in research since it is dealing with patient data.
Response 21: See response 9.
Comment 22º: The number of references is scarce for a subject as important as the one addressed by the authors.
Response 22: The number of references has been increased.

Reviewer 4 Report
Dear Authors,
I was pleased to review the paper entitled “Analysis and report of the Physical and Rehabilitation Medicine evaluation activity in patients admitted to acute care setting. An observational retrospective study”.
The topic is very interesting, in fact the acute Rehabilitation represent an important step for the orthopedic treatment. As this work confirms, the most frequently occurring setting for orthopedic rehabilitation is undoubtedly acute rehabilitation.
Therefore, it is my opinion that the content is current and relevant.
As regards the introduction and materials and methods, the sections are well described. In the introduction, I only suggest to spend a few words on the guidelines.
In result, when we analyzed the most frequently-occurring pathologies and that the most frquent was femoral neck fracture, we should underline its epidemiological and socioeconomic impact and the importance of the drug therapy in order to prevent this disease. So, in my opinion, we can enrich this section by quoting the following interesting scientific work:
- European review for medical and pharmacological sciences 26(1), pp. 43-52
-
The effect of combined drug therapy in lateral fragility fractures of the femur: a prospective observational study, Maccagnano G. et al.
As regards the discussion and conclusions, are balanced and well supported by analysis. In the conclusions i say a few word about how to reduce the time between hospitalitation and requeste for PRM.
As last tip, io would have done a multicentre study.
Author Response
Response to Reviewer 4 Comments
Dear Authors,
I was pleased to review the paper entitled “Analysis and report of the Physical and Rehabilitation Medicine evaluation activity in patients admitted to acute care setting. An observational retrospective study”.
The topic is very interesting, in fact the acute Rehabilitation represent an important step for the orthopedic treatment. As this work confirms, the most frequently occurring setting for orthopedic rehabilitation is undoubtedly acute rehabilitation.
Therefore, it is my opinion that the content is current and relevant.
As regards the introduction and materials and methods, the sections are well described. In the introduction, I only suggest to spend a few words on the guidelines.
Response 1: Thank you for your suggestion. We improved the introduction section explaining the national guidelines in terms of rehabilitation services and the need for early rehabilitation to guarantee better clinical outcome. We have added references about early rehabilitation to improve the patient's outcome.
In result, when we analyzed the most frequently-occurring pathologies and that the most frquent was femoral neck fracture, we should underline its epidemiological and socioeconomic impact and the importance of the drug therapy in order to prevent this disease. So, in my opinion, we can enrich this section by quoting the following interesting scientific work:
- European review for medical and pharmacological sciences26(1), pp. 43-52
● The effect of combined drug therapy in lateral fragility fractures of the femur: a prospective observational study, Maccagnano G. et al.
Response 2: Thank you for your suggestion, this is a very interesting observation. We integrated the discussion with a sentence regarding prevention strategies for fragility fractures adding the reference of the indicated article.
As regards the discussion and conclusions, are balanced and well supported by analysis. In the conclusions i say a few word about how to reduce the time between hospitalitation and requeste for PRM.
Response 3: we integrated the conclusion section as follows: “It is crucial to ensure a timely global taking care of the patient, through an early collaboration between the acute ward and the PRM specialists, to improve the outcome and reduce disability”.
As last tip, io would have done a multicentre study.
Response 4: thank you for the suggestion. This study is not multicentric, thus we indicated this as a limitation of our study.

Round 2
Reviewer 3 Report
Dear Authors.
The indications/recommendations indicated in the previous review have been partially solved.
- The main limitation of the study is the approval by a bioethics committee in research, with the generation of an identification code and approval of the study, we are dealing with a study where medical data are being collected from patients and where an analysis of these data is being performed, which requires this type of approval by a bioethics committee.
Informed consent is a mandatory requirement in any research process where research is performed on humans or data is collected from them, a department head report is not valid.
- Serious errors continue to be detected with the English language; no expert review report is attached.
- The introduction should have a greater scientific rigor and a better contextualization of the problem, since it is a complex subject that requires a more detailed and exhaustive analysis. It is not a good practice to carry out the introduction when more than 50% of its content is obtained from a single reference [ref. 9], which shows that an in-depth analysis has not been carried out.
- You did not determine the selection criteria, this carries a great risk of bias, you must clearly define the study population, both inclusion and exclusion criteria.
- You also failed to determine the sample size; this is also considered a big mistake.
- The authors were told: "The discussion section should elaborate on the clinical implications that may arise as a consequence of their results". They have not indicated any changes and expanded on the evidence regarding the subject of the study. They could provide information relevant to the repercussions that may arise from their results and how it may influence current physical rehabilitation, and contextualize the existing problem with relevant evidence.
Author Response
- The main limitation of the study is the approval by a bioethics committee in research, with the generation of an identification code and approval of the study, we are dealing with a study where medical data are being collected from patients and where an analysis of these data is being performed, which requires this type of approval by a bioethics committee.
We already had a response from the editor. Furthermore I would like to underline how, before submitting to the journal, we wrote to the editor specifying every detail of the study and the assistant editor Mark Szepesi replied to us on January 31, 2023, approving our specifications.
- Serious errors continue to be detected with the English language; no expert review report is attached.
One of the authors has a Bechelor degree in medicine in English recognized in every English-speaking country (we will attach a degree if necessary). If the reviewer noticed any English language errors please point them out correctly.
- The introduction should have a greater scientific rigor and a better contextualization of the problem, since it is a complex subject that requires a more detailed and exhaustive analysis. It is not a good practice to carry out the introduction when more than 50% of its content is obtained from a single reference [ref. 9], which shows that an in-depth analysis has not been carried out.
Thank you for the suggest, we improved the introduction section providing further evidence about this topic.
- You did not determine the selection criteria, this carries a great risk of bias, you must clearly define the study population, both inclusion and exclusion criteria.
As already widely underlined, our research is an observational study and all patients who underwent physiatric consultation were included without any exclusion or sample selection.
- You also failed to determine the sample size; this is also considered a big mistake.
See answer above.
- The authors were told: "The discussion section should elaborate on the clinical implications that may arise as a consequence of their results". They have not indicated any changes and expanded on the evidence regarding the subject of the study. They could provide information relevant to the repercussions that may arise from their results and how it may influence current physical rehabilitation, and contextualize the existing problem with relevant evidence.
We modify the text as follow: "This is the first Italian study conducted on a large scale in relation to physiatric medical consultations. Having been conducted in the largest hospital in Italy, it is likely to expect that these results could be useful at an international level to better understand the type of patients who need physiatric consultation and what are the specific characteristics that guide the physiatrist in expressing the rehabilitative diagnosis and prognosis".